# Intervention Programs for the Problematic Use of the Internet and Technological Devices: A Systematic Review

Elizabeth Cañas * and Estefanía Estévez 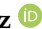

Department of Health Psychology, Universidad Miguel Hernández de Elche, 03202 Elche, Spain; eestevez@umh.es
* Correspondence: ecanas@goumh.umh.es; Tel.: +34-965-919-466

**Abstract:** The intensive use of the Internet and communication technologies among adolescents has increased addiction and/or their problematic use. The innovative and revolutionary development of this technology can have negative effects on the mental and physical health of its users, and it seems to have a greater impact on adolescents. As this is causing a public health problem, the objective of this study was to review the different intervention and prevention programs for this problem in adolescents. A total of 14 programs met the inclusion criteria. The analysis of the programs allows for the identification of effective intervention designs for prevention, and also for the treatment of the current problems derived from the use of the Internet and technological devices among adolescent users.

**Keywords:** intervention programs; Internet use; technological device use; adolescence





## 1. Introduction

The fast development of the Internet, as well as the intensive and continuous use of new information and communication technologies (ICTs), has provided numerous ways of communicating and interacting with others for the general population. However, the innovative and revolutionary development of these technologies seems to have a greater impact on adolescents [1].

The greater accessibility of the new communication media has increased the addiction and/or problematic use rates of the Internet, making it a potential health problem in today's society, and for which there is an increasing amount of research [2]. Internet addiction is characterized by excessive or poorly controlled preoccupations, urges, or behaviors with regard to computer use and Internet access that leads to impairment or distress, and that may indeed interfere with the daily functioning of individuals. This behavior can include addiction to online video games, technological devices, and social networks [3]. Although the DSM-V does not yet have a specific classification for this addiction, it has classified it within "Substance abuse and addiction disorders", as it is a compulsive behavior without substance abuse [4]. Evidence suggests that people who make problematic use of the Internet have a brain structure similar to people with dependencies on chemical substances (e.g., drugs or alcohol), which is reflected in the alteration of the prefrontal cortex [5]. Therefore, Internet addiction and/or problematic Internet use, like other dependency disorders, affect the brain's pleasure center, causing the release of dopamine. These chemical changes cause increasingly more activity and/or time needed to induce the same pleasant response, creating a dependency [6].

The problematic use of the Internet and new technologies can have negative effects on the mental and physical health of its users [7]. Some of the problems associated with these behaviors are loneliness and social isolation, aggression, anxiety [8,9], headache [10], and sleep disorders [11], among other somatic symptoms [12]. In addition, other problems, such as a lack of concentration, memory loss, fatigue, and stress have been reported [13,14]. Previous studies have recognized that Internet and mobile phone addiction have negative

effects, such as interpersonal problems [15–17], depressive disorders [18–25], lower life satisfaction [18,21], and conduct disorders, such as substance and alcohol abuse [23,26].

The consequences associated with the problematic use of the Internet and new technologies have been reported to a greater extent among adolescents [27]. This is because, on the one hand, and as mentioned above, adolescents make greater use of these new devices and spend more time connected to the Internet. Compared to adults, adolescents are more vulnerable to the problematic use of these media, as they are considered indispensable [28,29], leading them to experience a complex two-way relationship between what happens offline and what happens online. According to the co-construction model, adolescents shape their reality by connecting their offline and online worlds, the latter being the "dominant world" [30]. Children and adolescents are abandoning traditional leisure (e.g., playing outdoors, reading) and are replacing it with leisure through new media and technological devices [31]. On the other hand, adolescence is the stage of development where there is a greater risk of emotional crises, which, together with the excess amount of time invested in the online world, may be accompanied by mood swings, anxiety, and depression [31].

Access to ICTs has also become a precondition for the educational and occupational advancement of young people [6]. In this sense, the Internet can be a beneficial tool when used for educational purposes, such as preparing homework and acquiring knowledge. However, in most cases, adolescents use the Internet to make friends, communicate with others, and to play, rather than for academic purposes [32]. In a study carried out with a sample of 317,443 adolescents from 52 countries, López-Bueno, Koyanagi, López-Sánchez, Firth, and Smith, [33] observed that 66.9% of the sample made intense use of the Internet during weekdays, and 77.1% during weekends, not including school hours, and 23.2% of the adolescents reported intense Internet use at school [33]. This excessive use, together with the psychological immaturity typical of adolescence, turns youths into a group that is at high risk for Internet addiction, a fact that advocates for the design of prevention programs for this group [34].

Considering the plurality of concurrences and risks associated with the use of the Internet, their prevention and intervention are essential to promoting the positive use of technological media among young people. Although addiction and the problematic use of the Internet have begun to arouse great social interest, there are still few studies that focus on the prevention and intervention of these problems. Different health professionals and educators agree that treatment strategies for addressing the problem of Internet addiction must be accompanied by prevention strategies that include the risk factors before the addiction evolves into a more serious form [35,36]. Current scientific consensus calls for the development of well-controlled and methodologically solid interventions for the prevention and treatment of the problematic use of the Internet and new technologies that are based on empirical and theoretical evidence [6]. Among the most widely used programs for the intervention of this problem, we highlight cognitive behavioral therapy, which attempts to influence behaviors by reconstructing thoughts and feelings [37,38]. Another approach to addressing the problematic and addictive use of new technologies is multifamily therapy. This therapy relies on various mechanisms for intervening in this problem, such as improving communication and family cohesion, as well as satisfying the basic psychological needs of adolescents. These mechanisms can act as driving forces to promote behavioral change [39]. Positive psychology has also been one of the approaches used for treating the problematic use of the Internet and new technologies. Treatment focused on positive psychology is committed to increasing the frequency and quality of social contacts [40] and enhancing positive emotions [41], which could counter the maladjusted social relationships associated with the addiction to new technologies.

Since the terms "Internet addiction" or "problematic Internet use" first appeared in the mass media and the academic literature, research studies have gone a long way towards defining, exploring, investigating, describing, and predicting the phenomenon. Even so, the question, "How should Internet addiction be treated?", is still difficult to answer. Can

stopping all Internet usage cure Internet addiction? Many people believe the only way to cure Internet addiction is to stop using the Internet, and to unplug or to throw out the computer. Nevertheless, there are other recommended models for dealing with the misuse of, and addiction to, the Internet and new technologies. Approaches to this problem during adolescence are complicated because of the limited motivation for treatment and the therapy avoidance that teens show. Because of this, the school system is increasingly being used as a place to advance prevention efforts and to address health promotion and public health problems. Programs applied at school, or that cover a large number of adolescents, seem to be more efficient at tackling the various problems at this stage of development [42]. This study aims to expand the knowledge about the programs applied so far. Therefore, the objective of this study was to review the relevant literature on the programs that have been applied for the prevention and intervention of the problematic use of the Internet and new technologies in groups of adolescents, published between 2011 and 2021. The contribution of this article is based on the fact that it uses the results obtained by each of the programs reviewed as a guide.

## 2. Methods

The review was prepared following the PRISMA guidelines, for which definitions have been adopted from the Cochrane Collaboration [43]. The purpose of these guidelines is to ensure that the articles included are reviewed in their entirety, clearly and transparently. Figure 1 shows the flow diagram with the four phases recommended by the PRISMA guidelines. In the first screening, articles that had titles and abstracts that were irrelevant for this review, as well as duplicate articles, were excluded from the review. The last screening excluded articles that did not comply with the inclusion criteria, which are detailed in the figure legend.

### 2.1. Search Strategy

A systematic search was carried out for materials published in the last 10 years (from 2011 to the present) because, in this period, there was an exponential increase in digital media, as well as in social media platforms. The consultation was carried out through the following electronic databases: PsychInfo, Scopus, PubMed, and Web of Science. The search strategy was developed for each database using a combination of the terms: (problematic internet us * OR "compulsive internet us * OR pathological internet us * OR excessive computer us * OR internet addiction OR smartphone addiction OR cellphone addiction OR excessive mobile phone us * OR problematic smartphone us * OR problematic mobile phone us * OR problematic social media us * OR problematic social networking us * OR "social network addiction) AND (prevention program * OR intervention OR prevention strategy * OR treatment OR psychoeducation) AND (teen * OR adolescen *). Initially, duplicates were removed from the total number of records identified. The abstracts of the remaining references were screened to retrieve full-text manuscripts. Finally, studies that met the inclusion criteria were selected for evaluation.

### 2.2. Inclusion and Exclusion Criteria

The search was limited according to the following inclusion criteria:

(1) Studies on programs for the treatment and prevention of the problematic or addictive use of the Internet or different technological media;
(2) Studies whose objective (at least one) was to analyze the effectiveness of these programs;
(3) Studies in which the participants were adolescents enrolled in middle and secondary education, or secondary education centers;
(4) Quantitative or qualitative studies with cross-sectional or longitudinal designs;
(5) Articles in Spanish or English, because of the difficulties in translating articles in other languages.

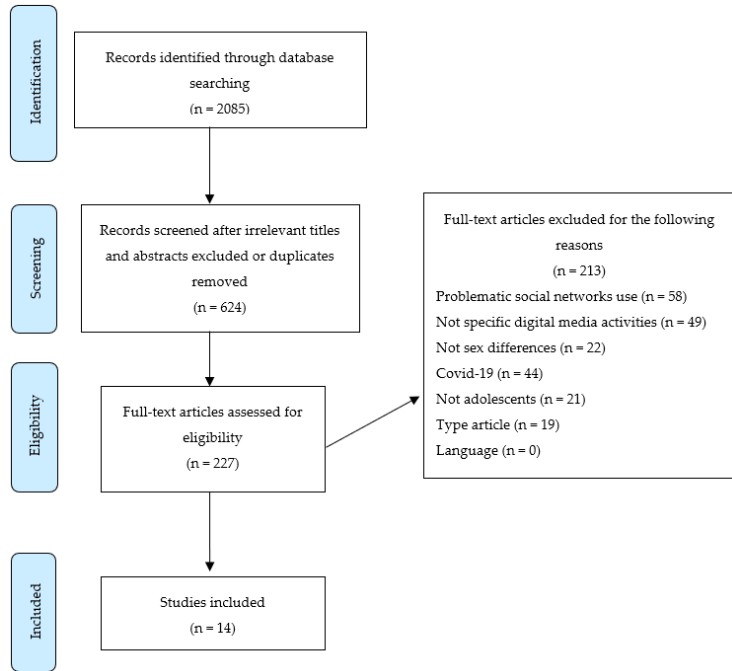

**Figure 1.** PRISMA flow of information through the different phases of a systematic review. Exclusion criteria: Problematic social network use: studies investigating the excessive use of, or addiction to, social media; Nonspecific digital media activities: studies not investigating specific activities on the Internet or different social media platforms; No sex differences: studies not analyzing gender differences; COVID-19: Studies analyzing the effects of social media use on mental health exclusively during the COVID-19 pandemic; Not adolescents: studies not involving adolescent participants; Type of article: nonquantitative studies or scientific articles; Language: studies not written in English or Spanish.

The exclusion criteria contemplated in the search were:

(1) Studies whose objective was not to treat or prevent the problematic or addictive use of the Internet or different technological media;
(2) Studies in which the applied program was not defined and detailed;
(3) Studies involving elementary and university students or adults;
(4) Reviews, editorials, theoretical articles, gray literature, dissertations, books, case studies, and conference proceedings without conference documents available in the databases;
(5) Articles in languages other than Spanish and English.

*2.3. Study Selection Process*

After compiling the manuscripts, we classified the studies, identifying those that met the inclusion criteria. For each of the studies, we extracted the following information: author and year of publication, study methodology, sample information, instruments for collecting data, key findings, and conclusions. These data were extracted by a researcher and verified by a second researcher to ensure the quality and accuracy of the information. Doubts or disagreements between evaluators were resolved through discussion and consensus with the help of a third reviewer. The results of this selection process are reported below.

**3. Results**

Broadly, the 14 studies included in this review can be divided according to the theoretical frameworks in which they developed their interventions for preventing and treating addiction to the Internet and new communication technologies. Thus, two programs were approached from cognitive behavioral therapy, nine from educational therapy, two from positive psychology, and one from multifamily therapy. The main results of the programs analyzed are described below, and Table 1 provides more information about them.

**Table 1.** Summary of selected studies.

| | Authors | Year | Participants | Intervention $A = \pi r^2$ | Program | Statistical Analysis | Results on Efficacy |
|---|---|---|---|---|---|---|---|
| Cognitive-behavioral therapy (CBT) | Du, Jiang, and Vance, [44]. | 2010 | 56 adolescents Exp. group: 32 Control group: 24 8 sessions China | Internet addiction | Topics discussed: 1. How to recognize and control your feelings; 2. Principles of healthy communication between parents and children; 3. Techniques for dealing with relationships developed via the Internet; 4. Techniques for dealing with content experienced via the Internet; 5. Techniques for controlling your impulses; 6. Techniques for recognizing when addictive behavior is occurring; 7. How to stop addictive behavior. The last session was a review session. | Paired *t*-tests comparing means (*M*) and standard deviations (*SD*) of each group. | Multimodal school-based group CBT is effective for adolescents with Internet addiction, particularly for improving emotional states and regulation abilities, and behavioral and self-management styles. |
| | Bong, Won, and Choi, [45]. | 2021 | 155 adolescents Exp. group: 70 Control group: 85 8 sessions Korea | Internet addiction Smartphone addiction | Music therapy 1. Self-introduction using songs and various types of percussion, creating a rhythm; 2. Personal history narration with games; 3. The participants drew upon their cravings for the games, explored their emotions, and checked external and internal factors through lyrics, rhythmic chanting, and playing instruments; 4. To think about the consequences of behavioral patterns, express emotions, and form empathy with them, and explore the reasons and motivations for game control on their own by working on rap lyrics; 5. Conducting and performing percussion music to improve their leadership and their ability to control their behavior; 6 and 7: Creating songs together using worksheets, and performing using sheet music that the therapist produced; 8. The notes of the first seven sessions were recorded in files that were presented as gifts to the participants. CBT was conducted by writing a home-based daily journal to evaluate the use of smartphones. | Paired *t*-test and analysis of covariance (ANCOVA) were used to compare the mean pre- and postintervention scores between groups. | CBT is effective in smartphone/Internet addiction, but its effectiveness increases when CBT is combined with music therapy |

**Table 1.** *Cont.*

| | Authors | Year | Participants | Intervention $A = \pi r^2$ | Program | Statistical Analysis | Results on Efficacy |
|---|---|---|---|---|---|---|---|
| Educational intervention | Yang and Kim, [46]. | 2018 | 79 adolescents Exp. group: 38 Control group: 41 10 sessions Korea | Internet addiction Internet usage time | Self-regulatory efficacy improvement program: 1. Introducing the program and building rapport; 2. Using the Internet properly; 3. Analyzing Internet addiction using self-inspection and setting goals for Internet use; 4. Improving self-cognitive, emotional, and behavioral control techniques; 5. Managing time effectively; 6. Relieving stress; 7. Managing physical and mental health; 8. Improving interpersonal relationships; 9. Finding alternative activities; 10. Designing the future. | Chi-squared tests, Fisher's exact tests, and independent $t$-tests to determine addiction scores and time spent on the Internet between groups. ANCOVA and paired $t$-tests to analyze intervention effects | Educational program led by school nurses who integrated and applied self-efficacy and self-regulation intervention strategies. Proved effective for the prevention of the students' Internet addictions. |
| | Gholamian, Shahnazi, and Hassanzadeh, [47] | 2019 | 120 adolescents Exp. group: 60 Control group: 60 2 sessions Iran | Internet addiction | The intervention followed BASNEF model constructs: knowledge, attitude, subjective norms, and enabling factors to reduce Internet addiction. | Chi-squared test was used to compare frequency of Internet use between groups. Mann–Whitney U test was also used to compare the duration of Internet use pre- and postintervention. Paired $t$-test was used to compare the mean scores of the BASNEF construct in each group. | The BASNEF constructs are a suitable framework for designing the educational interventions to reduce the extreme use of the Internet in students. |
| | Mathew, Krishnan, and Bhaskar, [6] | 2020 | 100 adolescents 12 weeks India | Problematic Internet use | Nurse-led intervention based on four modules and sub-modules: (a) Psychoeducation on problematic Internet use and its causes, incidence, prevalence, prevention, and management (one session); (b) Motivational interview (two sessions); (c) Behavioral modifications (three sessions); (d) Life skills to manage psychological problems, improve social skills, and prepare for academic examinations (six sessions). Two sessions were provided for parents on family psychoeducation and relapse prevention. | ANOVA was applied to examine the overall effectiveness of the intervention. Post hoc analysis with Tukey's HSD correction was conducted for both groups to analyze the differences for PIU at three time periods: T1 (baseline), T2 (Week 1), and T3 (Week 12). | Nurse-led intervention is effective in reducing problematic Internet use among adolescents and it improved their physical, psychological, and social functioning and academic performance. |

**Table 1.** *Cont.*

| Authors | Year | Participants | Intervention $A = \pi r^2$ | Program | Statistical Analysis | Results on Efficacy |
|---|---|---|---|---|---|---|
| Çelik, [34] | 2016 | 30 adolescents<br>Exp. group: 15<br>Control group: 15<br>5 sessions<br>Turkey | Problematic Internet use | Training program sessions to increase conscious Internet use, academic motivation, and efficient use of time<br>1. Conscious use of the Internet;<br>2. Effective use of time;<br>3. Academic motivation;<br>4. Productive study;<br>5. Assessment. | ANOVA was used to determine whether Internet addiction was affected by the training program. | The educational program, developed to increase conscious Internet use, academic motivation, and efficient use of time, is effective in reducing the Internet addiction tendencies of adolescents. |
| Uysal, and Balci, [48] | 2018 | 64 adolescents<br>Exp. group: 32<br>Control group: 32<br>8 sessions<br>Turkey | Internet addiction | Healthy Internet Use Program included:<br>1. Self-recognition;<br>2. Self-expression;<br>3. Healthy Internet use;<br>4. The effects and problems of Internet addiction on social life, and of a sedentary lifestyle on Internet use;<br>5. Introduction of proper methods for encouraging healthy Internet use and coping with Internet addiction;<br>6. Creating awareness to set goals and achieve changes in behavior. | Chi-squared test was used to examine the relationship between variables.<br>Mann–Whitney U test was used to compare groups. | The Healthy Internet Use Program decreases the rate of Internet addiction among adolescents. |
| Ortega-Barón, González-Cabrera, Machimbarrena, and Montiel, [49] | 2021 | 165 adolescents<br>Exp. group: 120<br>Control group: 45<br>16 sessions<br>Spain | Cyberbullying, Sexting, Online grooming, Cyberdating abuse, Problematic Internet use, Nomophobia, Internet gaming disorder, Online gambling disorder | The Safety.net program was composed of four modules:<br>(1) Digital skills: to educate about the characteristics and risks of technologies and provide skills to prevent the dysfunctional use of the Internet;<br>(2) Relational risks: to raise awareness about the seriousness of the risks arising from the Internet and to advise about these problems;<br>(3) Dysfunctional risks: to become aware of the seriousness of the risks derived from the dysfunctional use of the Internet and provide advice on how to safely use ICT's;<br>(4) Change of attitudes and cognitions: to promote certain skills, competencies, and abilities to cope with the risks of the Internet. | ANOVA was used with an inter-group factor (intervention group and control group) and an intra-subject factor (before and after the program: pretest and post-test). | The Safety.net program is effective in preventing the increase in most of the assessed risks, and it reduces some of them with a small number of sessions. |

**Table 1.** *Cont.*

| Authors | Year | Participants | Intervention $A = \pi r^2$ | Program | Statistical Analysis | Results on Efficacy |
|---|---|---|---|---|---|---|
| Khoshgoftar, Amidi Mazaheri, and Tarrahi, [7] | 2019 | 112 adolescents Exp. group: 56 Control group: 56 6 sessions Iran | Mobile phone addiction | Educational intervention based on the health belief model (HBM): 1. To perceive susceptibility and to increase knowledge about mobile phone addiction; 2. To discuss the perceived severity, and short- and long-term negative consequences of mobile phone addiction; 3. To explore perceived benefits of, and barriers to, proper mobile phone usage; 4. To promote self-efficacy; 5. To develop an attractive signal about the correct use of the phone; 6. To review and summarize the subjects of the previous sessions. | Paired *t*-test was used to evaluate the effectiveness of interventions in each group. | Educational intervention based on the HBM prevents and decreases mobile phone addiction in female students. |
| Szász-Janocha, Vonderlin, and Lindenberg, [50] | 2020 | 54 adolescents 4 sessions Germany | Internet use disorders (IUDs) including: Internet gaming disorder (IGD) and nongaming pathological Internet use (ng-PIU) | PROTECT+ intervention and prevention: 1. Handling of boredom and motivational problems; 2. Reduction in procrastination and performance anxiety; 3. Reduction in social anxiety and promotion of social skills; 4. Promotion of functional emotion regulation skills. | Growth models in a hierarchical linear model framework to analyze intent-to-treat. | The results indicate that even a brief 4-session PROTECT+ intervention can achieve a medium-to-large effect over 12 months. |
| Walther, Hanewinkel, and Morgenstern, [51] | 2014 | 1843 adolescents Exp. group: 804 Control group: 1039 4 sessions Germany | Problematic Internet use, problematic computer game use | Vernetzte www.Welten ("connected www.worlds"): 1. Internet use; 2. Online communication; 3. Gaming; 4. Gambling. The focus lies on self-monitoring, discussion, and reflection on adolescents' media use. | Baseline differences between the control group and the intervention group were tested using chi-squared and *t*-tests. To test intervention effects over time, multilevel growth curve models with maximum likelihood estimations were applied. | Program Vernetzte www.Welten is effective in influencing the media-use behavior of adolescents. |

**Table 1.** *Cont.*

| | Authors | Year | Participants | Intervention $A = \pi r^2$ | Program | Statistical Analysis | Results on Efficacy |
|---|---|---|---|---|---|---|---|
| Positive Psychology | Khazaei, Khazaei, and Ghanbarih-H, [52]. | 2017 | 48 adolescents Exp. group: 24 Control group: 24 10 sessions Iran | Internet addiction | Positive psychology interventions: 1. To explain the assumptions and the role of positive psychotherapists in the treatment; 2. To review positive stories and to identify capabilities and strengths in the stories; 3. To formulate a specific plan for implementing abilities; 4. To understand the role of positive and negative memories in maintaining the symptoms of depression; 5. To focus on forgiveness; 6. To focus on thankfulness; 7. To review treatment progress; 8. To teach satisfaction versus perfectionism to subjects; 9 and 10. To focus on hope and optimism. | Inferential statistics, including one-way and multiway ANCOVA were used to control the effect of preintervention. | Positive psychology is an effective method for treating Internet addiction, specifically in mitigating Internet use and improving the quality of social relationships. |
| | Ke and Wong, [53] | 2017 | 157 adolescents 8 sessions Malaysia | Problematic Internet use | Program based on cognitive behavioral therapy incorporating positive psychological techniques. It is categorized into three modules: (1) Formulation; (2) Restructuring of thoughts and behaviors; (3) Modification of thoughts and behaviors. | Paired samples *t*-test was employed to determine whether there were significant changes in symptoms during the three phases of the program. | Problematic Internet use and related symptoms decreased after the eight weekly intervention sessions, showing the efficacy of this intervention program. |
| Multi-family group therapy | Liu et al. [1] | 2014 | 46 adolescents 46 parents 6 sessions China | Internet addiction | During the multifamily group therapy sessions, the following topics were discussed: 1. Understanding a family with an Internet addict; 2. Parent–adolescent communication skills training; 2 and 3. Parent –adolescent communication practices on Internet addiction; 4. Parent–adolescent skill-building relationship training; 5. Associations between psychological needs and Internet use and how to satisfy the unfulfilled needs in the family relationships; 6. Setting up appropriate and healthy expectations for the family system. | *t*-tests and repeated measures ANOVA analyses were conducted to determine the effectiveness of the intervention through the comparison of adolescents' Internet addiction measures in the intervention and control groups at T1, T2, and T3. | Multifamily group therapy is effective in reducing Internet addiction behaviors among adolescents and can be implemented as part of routine primary care clinical services in similar populations. |

### 3.1. Programs Based on Cognitive Behavioral Therapy

Among the studies that analyzed the effect of cognitive behavioral therapy, we highlight the work of Du et al. [44], whose results reveal that participants in the experimental group reported higher mean scores on some subscales related to excessive Internet use, such as "time effectiveness" and "control over time", immediately after the intervention (M = 33.6 and M = 34.48, respectively) and at 6 months of follow-up (M = 76.08 and M = 76.09, respectively) compared with the mean scores reported before the intervention (M = 28.21 and M = 64.95, respectively). On the contrary, the control group did not differ in any of the subscales analyzed.

In contrast, the study by Bong et al. [45] combined the intervention based on cognitive behavioral theory with music therapy. These authors observed significant decreases in the scores of the addictions to smartphones and the Internet after the cognitive behavioral program and the program combining cognitive behavioral therapy and music therapy. However, the scores were different depending on the intervention. Thus, the mean scores in the cognitive behavioral therapy/music therapy group dropped from 32.75 to 26.48 ($t = -5.880$, $p < 0.001$) for smartphone addiction, and from 46.79 to 32.16 ($t = -7.373$, $p < 0.001$) for Internet addiction. In the group in which only cognitive behavioral therapy was applied, the mean scores of smartphone addiction decreased from 33.42 to 28.65 ($t = -6.300$, $p < 0.001$), and Internet addiction dropped from 46.24 to 37.61 ($t = -4.124$, $p < 0.001$). With regard to the differences in the mean values before and after the intervention between the groups, significant differences were only observed in the magnitude of the reductions on the Internet Addiction Scale, and they were higher for the cognitive behavioral therapy/music therapy group ($Z = 5.137$, $p = 0.025$).

### 3.2. Programs Based on Educational Intervention

Gholamian et al. [47] observed no significant differences between the control and experimental groups in excessive Internet use before the intervention ($p = 0.68$). However, after the intervention, the intensity of Internet use was significantly lower in the experimental group than in the control group ($p < 0.001$). These authors observed that the scores on the program factors aimed at reducing Internet addiction before the intervention did not show significant group differences ($p > 0.05$). However, after applying the intervention, the mean scores of the above factors in the intervention group showed a significant increase compared to the control group ($p < 0.001$), except for the factor, "positive attitude towards the Internet", which was significantly lower than before the intervention ($p < 0.001$). Yang and Kim's [46] program showed significant differences between the control and experimental groups in the Internet addiction mean scores after the intervention ($M_{pretreatment}$ = 80.96 and 82.42, respectively, and $M_{posttreatment}$ = 85.37 and 61.13, respectively, $p < 0.001$). These authors also found significant differences between the control and experimental groups in the mean times (minutes/day) of Internet use, both for the weekdays ($M_{pretreatment}$ = 151.95 and 112.11, respectively, and $M_{posttreatment}$ = 209.02 and 98.68, respectively, $p < 0.001$) and the weekends ($M_{pretreatment}$ = 372.44 and 290.53, respectively, and $M_{posttreatment}$ = 392.68 and 164.74, respectively, $p < 0.001$), which suggests that the program had a significant influence on Internet addiction, as well as on the time of Internet use.

Among the studies that took longitudinal measures, the educational intervention led by school nurses in the study by Mathew et al. [6] had a significant effect ($p < 0.01$) on the mean scores of the experimental group in problematic Internet use in the three time periods (baseline, immediately after treatment, and 12 weeks later) in which the measures were collected. The results show higher mean scores in the control group than in the experimental group at these three periods ($M_{pretest}$ = 61.27 vs. 52.13; $M_{posttest\ 1}$ = 60.93 vs. 34.33; $M_{posttest\ 2}$ = 62.03 vs. 42.57, respectively). Çelik's [34] intervention showed a decrease in the mean scores of the intervention group compared to the control group after applying the intervention ($M_{pretest}$ = 114.56 vs. 109.93; $M_{posttest}$ = 89.06 vs. 107.31, respectively), and six months after the intervention ($M_{follow-up\ Test}$ = 90.31 vs. 107.31). Similarly, Uysal and Balci [48] observed that, for the scores of the participants with Internet addiction, in the

experimental group compared to the control group, addiction showed a significant decrease at 3 months (M = 76.41 vs. 84.91, $p < 0.05$, respectively) and at 9 months (M = 72.59 vs. 88.28, respectively), compared to the baseline (M = 103.37 vs. 100.77, $p < 0.001$, respectively).

Regarding the use of technological devices and online/offline video games, Khoshgoftar et al. [7] compared the mobile phone addiction scores in the intervention and control groups before and after the application of the program. The mean of the intervention group decreased significantly after the intervention (M = 30.25 vs. M = 26.76, $p = 0.02$, respectively), while, in the control group, the scores were maintained (M = 29.19 vs. M = 31.96, $p < 0.01$, respectively). Along these lines, in the pilot study by Ortega-Barón et al. [48], the results show the significant effect of the intervention in the experimental group compared to the control group in the problematic use of the Internet ($M_{\text{pre-intervention}}$ = 15.26 vs. 14.41; $M_{\text{post-intervention}}$ = 16.71 vs. 25.83, $p < 0.05$, respectively) and in Internet gambling disorder ($M_{\text{pre-intervention}}$ = 3.24 vs. 2.68; $M_{\text{post-intervention}}$ = 3.88 vs. 5.56, $p < 0.05$, respectively). These problems were more visible in the control group than in the intervention group. At the same time, a significant interaction effect was found on the nomophobia variable ($F(1, 147) = 19.90$, $p = 0.01$), showing that, while the scores of the intervention group were lower, the control group's scores for this problem increased ($M_{\text{pre-intervention}}$ = 35.54 vs. 26.48; $M_{\text{post-intervention}}$ = 28.80 vs. 44.46, $p < 0.001$, respectively). However, no significant main or interaction effects were found for online gambling disorders.

Longitudinal studies, such as those of Szász-Janocha et al. [50], observed that the baseline scores for compulsive Internet use and dependence on video games ($M_{\text{pretest}}$ = 23.49 and M = 16.75, respectively) decreased at 4 months after applying the PROTECT + program, ($M_{\text{4 months follow-up}}$ = 21.45 and 14.58, respectively), and at 12 months ($M_{\text{12 months follow-up}}$ = 14.14 and 9.75, respectively). The dependence on video games reported by the parents showed a similar pattern ($M_{\text{pretest}}$ = 33.69 and $M_{\text{postest}}$ = 27.79), except for a small rebound at 4 months ($M_{\text{4 month follow-up}}$ = 29.72) that decreased again at 12 months ($M_{\text{12 month follow-up}}$ = 26.80), which corroborates the effectiveness of the program. In contrast, the program of Walther et al. [50] managed to reduce the frequency and duration of video games in the experimental group compared to the control group from the beginning of the intervention (T1) ($M_{\text{frequency}}$ = 8.14 vs. 8.36, respectively, and $M_{\text{duration}}$ = 1.06 vs. 0.95, respectively) until immediately after its application (T2) ($M_{\text{frequency}}$ = 6.96 vs. 7.80, respectively, and $M_{\text{duration}}$ = 0.99 vs. 1.06, respectively). However, these authors report an increase in both measures at follow-up (T3) in both groups (experimental and control groups) ($M_{\text{frequency}}$ = 8.04 and 9.35, respectively; $M_{\text{duration}}$ = 1.18 and 1.29, respectively). The frequencies and durations of Internet use in the experimental and control groups also increased over time, and the proportion of daily Internet users almost doubled within the study interval (15 months). However, except for excessive Internet use (every 4 h per day) and Internet addiction, where the mean scores were higher in the control group, there were no visible differences in the frequencies and durations between the two groups.

### 3.3. Programs based on Positive Psychology

Although the program applied by Ke and Wong [53] is a cognitive behavioral intervention, it incorporates positive psychology techniques. The results of this study show that, at the end of the intervention, the mean scores of the participants who had problematic Internet use were significantly lower than the mean scores obtained prior to the intervention (T0–T1) ($M_{\text{pre-intervention}}$ = 58.81 vs. $M_{\text{post-intervention}}$ = 45.38, $p < 0.01$). Postintervention and follow-up scores (T1–T2) also showed a significant decrease in problematic Internet use ($M_{\text{post-intervention}}$ = 45.38 vs. $M_{\text{follow-up}}$ = 37.22, $p < 0.01$), which implies that the program not only reduces problem behavior but that this is maintained over time. Overall, the results show that, after 8 weeks, most participants could differentiate between the healthy and unhealthy use, or excessive use, of the Internet. These results suggest that, after therapy, the participants could limit their Internet use and develop new social relationships over time. Similarly, in the study by Khazaei et al. [52], the mean scores of the experimental group were significantly reduced after the application of the positive

psychology intervention compared to the control group, in both the severity of Internet use ($M_{\text{pre-intervention}}$ = 60.3 vs. 59.3; $M_{\text{post-intervention}}$ = 12.6 vs. 62.4, $p < 0.01$, respectively) and Internet addiction ($M_{\text{pre-intervention}}$ = 63.5 vs. 63.2; $M_{\text{post-intervention}}$ = 47.2 vs. 64.1, $p < 0.01$, respectively).

*3.4. Programs Based on Multifamily Group Therapy*

In the program applied by Liu et al. [1], the scores related to pathological Internet use in the three time periods were significant ($p < 0.001$), and they also showed a linear decrease in time before the intervention (T1) ($M_{\text{experiemntal\_group}}$ = 3.40 vs. $M_{\text{control\_group}}$ = 3.38), compared to immediately after the intervention (T2) ($M_{\text{experiemntal\_group}}$ = 2.46 vs. $M_{\text{control\_group}}$ = 3.59), and up to 3 months later (T3) ($M_{\text{experiemntal\_group}}$ = 2.06 vs. $M_{\text{control\_group}}$ = 3.27), which indicates that the effects of the intervention were maintained over time. Time spent on the Internet also showed a significant decrease from T1 to T3 ($p < 0.001$) (Experimental group: $M_{\text{T1}}$ = 26.38, $M_{\text{T2}}$ = 11.43, and $M_{\text{T3}}$ = 7.08, vs. Control group: $M_{\text{T1}}$ = 27.08, $M_{\text{T2}}$ = 27.52, and $M_{\text{T3}}$ = 22.29. Moreover, at 3 months after the intervention, parents continued to report a significant reduction in their perception of their children's Internet addiction, revealing a significant linear decline from T1 to T3 ($p < 0.001$) (Experimental group: $M_{\text{T1}}$ = 3.37, $M_{\text{T2}}$ = 3.13, and $M_{\text{T3}}$ = 2.70, vs. Control group: $M_{\text{T1}}$ = 3.36, $M_{\text{T2}}$ = 3.45, and $M_{\text{T3}}$ = 3.20).

**4. Discussion**

The arrival of the Internet and new technological devices has changed and revolutionized ways of relating, overcoming the physical and temporal barriers of the "real world". The high rates of individuals who have incorporated this form of communication into their daily lives, especially adolescents, worries experts, given the problematic, and even addictive, use that has been related to these devices [54]. Therefore, the implementation of interventions aimed at preventing and treating this problem is essential to increasing the awareness of the negative effects of the use of the Internet and new technologies, as well as to promoting healthy lifestyle behaviors. The present work carried out a systematic review with the aim of providing a general description of the intervention programs available for treating addiction and the problematic use of the Internet and technological devices from a multidisciplinary perspective. Specifically, 14 studies were selected that have analyzed programs framed in various therapies, such as cognitive behavioral therapy, music therapy, educational therapy, positive psychology, and multifamily therapy. Each of them has been shown, to a greater or lesser extent, to be effective for intervening in the problems derived from the use of the Internet and technological devices.

*4.1. Programs Based on Cognitive Behavioral Therapy*

Studies that have applied cognitive behavioral programs as an intervention for problematic Internet use have aroused some controversy, as addictions to technological devices and the Internet are not considered mental illnesses. However, the evidence supports that psychological therapy is useful if addiction to the Internet and new technologies is severe enough to affect the daily lives of its users [55,56]. More specifically, cognitive behavioral therapy is widely studied and is currently the first line of treatment for addictions to technological devices. The effectiveness of this therapy is explained, either because it allows for the effective correction of nonadaptive automatic thoughts, or because it reduces the psychosocial symptoms related to the addictive experience [57,58]. Several researchers have pointed out that the benefits of cognitive behavioral therapy in the treatment of Internet addiction is maintained over time, especially when considering that the reduction in the time invested in connection to the Internet allows for the rekindling of offline social relationships that were lost as a result of Internet addiction [53].

Other works have gone one step further by combining cognitive behavioral therapy with music therapy. This is the case of the work of Bong et al. [45], who applied the synergistic interventions of both therapies as a treatment for Internet addiction, observing not only less problematic use of the Internet, but also fewer anxiety symptoms and less

impulsivity than when only cognitive behavioral therapy was applied. These authors argue that music therapy not only relaxes participants emotionally and allows them to experience intimacy with the therapist and with group members, but also encourages the performance of cognitive behavioral therapy tasks, in addition to inducing interest in the group therapy and adaptations to it. However, given the scarcity of results regarding the effectiveness of music therapy in addiction disorders, more studies are required to determine the effects of this therapy as a treatment for the problematic use of the Internet and new technologies.

### 4.2. Programs Based on Educational Intervention

Findings from other studies have shown the effectiveness of educational programs for the treatment of addiction to the Internet and new technologies. Most of these programs are supported by environmental factors, such as the school and its members, and cognitive factors, such as self-regulation and self-efficacy [59,60]. These two factors have strong effects on addiction. According to social cognitive theory [60], an inadequate capacity for self-regulation could lead to habitual Internet users not controlling their use. In this respect, self-efficacy, defined as the strength or level of belief in one's ability to complete tasks, attain success, and achieve objectives can be a key element in the modification of behavior in adolescence [59]. Studies carried out show that the benefits of self-efficacy at the behavioral level can be achieved, even indirectly. That is, classmates who have successful control over the use of technological devices can act as positive role models and promote perceived self-efficacy in controlling the use of these devices through vicarious learning [61].

The formative intervention of educational programs on the prevalence and negative consequences of Internet addiction also seems to have a beneficial impact on behavioral performance and Internet use [47]. As the problems derived from the time invested online reduce the motivation of the adolescent population towards fulfilling academic responsibilities [34], various educational programs have focused on developing academic motivation and increasing the efficient use of time, and have proved effective in reducing the trend of Internet addiction [62,63]. Most of these programs aim to teach efficient strategies for the management of time, as well as to encourage attention, motivation, and productive study. These objectives are fundamental for treating Internet addiction, especially among adolescents, because people with a predisposition to this problem have difficulties efficiently managing the time they spend on the Internet [64].

In addition to focusing on increasing motivation, it is also necessary to address other factors, such as raising awareness about the risks and consequences derived from the inappropriate use of the Internet and technological devices. Despite the fact that many young people are unaware of the harmful factors associated with new technologies, after intervention through educational programs, adolescents begin to become aware of the risks involved in their inappropriate, and even addictive, use [14,65]. Thus, the effectiveness of educational programs seems to be based on an awareness of the negative consequences of one's actions, which consequently increases people's motivation to adopt healthy behaviors [66] and to avoid addictive behaviors [59]. It has been found that programs that did not raise awareness about the severity of Internet addiction [66], or that did not provide practical opportunities for promoting self-control factors [67,68], did not show any significant effect on the time of Internet use.

However, the results of the programs may vary depending on the problematic use they are intended to address. Some investigations that have measured various problem behaviors, such as the use of video games and the Internet, have only found efficacy in one of the two behaviors. For example, in the case of Walther et al. [51], positive effects were only found in video games and not in the time of Internet use. The authors explain these findings through a question of measurement. While video games are a well-defined activity that can be easily reported, the use of the Internet is much more multifaceted, allowing for a wide variety of activities. Thus, quantifying Internet use can be a difficult task and, therefore, its evaluation is more prone to errors [69]. Another possible explanation is that

Internet use is routine in the daily lives of today's teenagers, and reducing the frequency or duration of its use is simply more difficult [70]. However, the design proposed by other programs has not only reduced the rates of Internet gambling disorders, but also those of problematic Internet use. For example, the study of Ortega-Barón et al. [49] not only proved to be an ideal strategy for primary prevention (avoiding the appearance of problems), but also for secondary prevention (preventing situations from worsening in the cases of adolescents who were already at a certain level of risk) for the target population. These results empirically confirm that such programs must include intervention strategies that consider the individual cognitive factors of adolescents, environmental strategies with their peers, and behavioral strategies that allow them to independently practice the activities they choose.

### 4.3. Programs Based on Positive Psychology

Programs based on positive psychology have also been effective in easing the intensity of online connections in people with Internet addiction problems. However, another favorable effect of these programs is related to the social maladjustment associated with Internet addiction. Therefore, positive psychology is not only effective in reducing the intensity of Internet use, but it also promotes social harmony and adaptation among individuals. Various works emphasize that the mission of positive psychology is to enhance human capacities, which is why it seeks to provide people with happier and healthier lives. These results were observed in the study by Khazaei et al. [52], in which they found that the quality of the relationships of students with Internet addictions who participated in a positive psychology intervention was significantly higher than in the control group. These authors point out that positive psychology interventions help the treatment of Internet addiction by promoting social adjustment and improving the quality of social relationships. The results show that positive psychological interventions in people with addictive behaviors can improve interpersonal relationships, which is associated with a reduction in the symptoms of the disorder [70,71]. In fact, there is a strong negative correlation between the severity of Internet addiction symptoms and the quality of interpersonal relationships such that, as symptoms ease, interpersonal relationships improve. Thus, as the destruction of social interactions by people with Internet addiction is inevitable, one of the main aspects of positive psychological treatment is to alleviate and preserve interpersonal relationships [52].

### 4.4. Programs Based on Multifamily Group Therapy

Studies focused on multifamily therapy show effectiveness in reducing the time spent on the Internet, which is maintained in the long term. Studies argue that improving communication between parents and adolescents, as well as learning alternative ways to meet the needs of young people, can reduce dependence on the Internet and promote the motivation to sustainably change their behavior. The satisfaction of needs in real life has been associated with a decrease in Internet addiction [52]. In addition, the literature highlights that adolescents who maintain positive relationships with their parents feel less dependence on the Internet, and it is easier for them to find alternatives to satisfy their needs [1,72]. The family system approach shifts the emphasis from individual members to the whole family as a unit, and the dynamic interactions between family members [73]. In fact, the participation of other family members in the intervention can create a more supportive environment in which the behavior changes of the participants are valued, encouraged, and maintained, even after the intervention ends [74]. The effects of multifamily therapy have been supported by the literature.

## 5. Conclusions

The findings of the analyzed studies corroborate the idea that intervention programs for adolescents are necessary, not only to prevent, but also to treat current problems derived from the use of the Internet and technological devices. In one way or another,

all the programs have focused on sensitizing people to, and raising awareness about, the risks derived from the excessive use of, and connection to, the Internet, which has made it possible to observe a generalized reduction in the maladaptive behaviors of the participants in the different studies. The programs also emphasized the consideration of not only the individual, but also the peer group, teachers, and family as part of the intervention, considering that the rest of the systems in the program increase the effectiveness of the treatment, and also its maintenance over time.

Although this review represents a relevant contribution to the existing scientific literature, it is not without limitations. The diversity of the programs analyzed makes it difficult to generalize the results. The variety of objectives, measurement instruments, and the designs of the interventions hinders comparisons between studies in terms of effectiveness. To these considerations can be added others, such as the population of the study, which was made up only of adolescents, and the search criteria, which included programs in which only one measure was taken, and others, in which there were follow-up measures. In addition, the context of the COVID-19 pandemic and the demands generated have changed society's dynamics concerning the use of the Internet. This change makes it necessary to understand Internet addiction in this new context in order to be able to generate effective diagnostic criteria and, subsequently, develop treatments that adequately respond to the particular conditions of this problem. Therefore, this work should be considered as a starting point for future research that incorporates programs that have been implemented and validated with students from other academic levels.

**Author Contributions:** Conceptualization, E.C.; methodology, E.C.; validation, E.C. and E.E.; formal analysis, E.C.; resources, E.C.; data curation, E.E.; writing—original draft preparation, E.C.; writing—review and editing, E.E.; visualization, E.C. and E.E.; supervision, E.E.; project administration, E.E.; funding acquisition, E.E. All authors have read and agreed to the published version of the manuscript.

**Funding:** This study is part of the project, "Bullying and cyberbullying among peers and in adolescent couples: from emotional regulation to suicidal"—Reference: PID2019-109442RB-I00, funded by the Ministry of Science, Innovation and Universities of Spain, with Estefanía Estévez as the Principal Researcher.

**Institutional Review Board Statement:** Not applicable.

**Informed Consent Statement:** Not applicable.

**Data Availability Statement:** Not applicable.

**Conflicts of Interest:** The authors declare no conflict of interest.

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
