# Peer review of "Intervention Programs for the Problematic Use of the Internet and Technological Devices: A Systematic Review"

_electronics, doi:10.3390/electronics10232923_

Round 1
Reviewer 1 Report
The paper presents a survey about prevention and intervention of the problematic use of the Internet.
The methodology of the compilation of the survey is very clear and correct (I appreciated the use of Cochrane Collaboration (please add reference in the article).
The discussion
There are two main issues with this paper:
- when you comment the papers of the survey, it would be helpful to explain the role of the "scores" in each study (for example, what are "scores in cognitive behavioural therapy"?). Also, I would suggest to give some hints about the magnitude of the scores.
- The authors should motivate the paper within the aims and scopes of the journal. I wonder why this article was not submitted to other MPDI journals like Psych or Life (they had a special issue related to this topic as well https://www.mdpi.com/journal/psych/special_issues/Internet_Addiction_Psych)
Minor issues:
- Can the number of paper of the survey be incremented by relaxing some of the constraints. This is just a suggestion, maybe the specific research area does not have too many papers (a bit more one per year given the timespan).
- please clarify what are Figure 1 and Table 1 in the paper (from the supplementary files it is not clear what is what)
Author Response
The paper presents a survey about prevention and intervention of the problematic use of the Internet.
The methodology of the compilation of the survey is very clear and correct (I appreciated the use of Cochrane Collaboration (please add the reference in the paper).
- We have added the Cochrane Collaboration reference.
The discussion
There are two main issues with this paper:
- when you comment the papers of the survey, it would be helpful to explain the role of the "scores" in each study (for example, what are "scores in cognitive behavioural therapy"?). Also, I would suggest to give some hints about the magnitude of the scores.
- We have edited the results section making the scores of each analyzed study more understandable.
- The authors should motivate the paper within the aims and scopes of the journal. I wonder why this article was not submitted to other MPDI journals like Psych or Life (they had a special issue related to this topic as well https://www.mdpi.com/journal/psych/special_issues/Internet_Addiction_Psych)
- Thanks for the observation. We are aware that there are many other journals more in line with the subject of this study, however, the editor contacted us to prepare an article for this special issue. We proposed this topic to him and it seemed like a good idea.
Minor issues:
- Can the number of paper of the survey be incremented by relaxing some of the constraints. This is just a suggestion, maybe the specific research area does not have too many papers (a bit more one per year given the timespan).
- We did not think that relax any limitation is appropriate since the stage of development (adolescence) where we focus is the stage in which the use of technological devices predominates and in which the consequences are very harmful.
On the other hand, we did not broaden the dates criteria, since technologies and social networks change very quickly. Programs intended for problem use from 10 years ago are probably not effective for today's teens. It can be seen that most of the studies are not older than 5 years (we have added a new column in Table 1 specifying the year of the study).
- please clarify what are Figure 1 and Table 1 in the paper (from the supplementary files it is not clear what is what).
- Thanks for the suggestion. We have edited Figure 1 and Table 1 and incorporated the paper to make it more understandable to the reader.
* We have used the translation service to correct possible errors in grammar and spelling.

Reviewer 2 Report
Intervention programs for the problematic use of the Internet and technological devices: A systematic review
General
In this paper, the authors focus on the review of 12 different intervention and prevention programs for the problematic use of the Internet and technological devices in adolescents. The analysis of the programs makes it possible to identify effective intervention designs for preventing but also to treat current problems derived from the use of the Internet and technological devices among adolescent users.
The work is informative and enjoyable to read, the results are convincing, and the paper is well-written, even if the presentation could benefit from minor reorganization and corrections.
Main comments
Here is a suggestion in which the paper could be improved to reach a broader audience.
#1 I think there are some missing tables and figures on page 4.
#2 There are a series of measures/scores that are not clear for a broader audience.
I suggest explaining the description of “M, t, p, Z, F(2), F, χ, F(1, 147), F(1, 157)” and how they are measured. The authors should at least name them explicitly.
For instance, the sentence
“(…) the scores referring to smartphone 185 addiction decreased from 33.42 ± 6,487 to 28.65 ± 7,968”.
How can a reader interpret this information? How is this score calculated? Is it counted in minutes or other quantities?
#3 The authors should assist the reader in understanding their results. The paper is well-written, but in Section 3, it became exceedingly technical. I do not recommend removing any data or result, but explaining the metrics to the reader would improve the overall quality of this work.
Overall, I think the authors should define/explain/name each metric presented in Section 3.
Recommendation
In my opinion, the scientific content of the paper is sound and the experimental results are convincing. The treatment of literature seems fair. Furthermore, I think that enough methodological details are provided for this work to be reproduced. Therefore, I recommend that this paper should be accepted for publication after minor revision.
Author Response
General
In this paper, the authors focus on the review of 12 different intervention and prevention programs for the problematic use of the Internet and technological devices in adolescents. The analysis of the programs makes it possible to identify effective intervention designs for preventing but also to treat current problems derived from the use of the Internet and technological devices among adolescent users.
The work is informative and enjoyable to read, the results are convincing, and the paper is well-written, even if the presentation could benefit from minor reorganization and corrections.
- Thank you very much for your comment.
Main comments
Here is a suggestion in which the paper could be improved to reach a broader audience.
#1 I think there are some missing tables and figures on page 4.
- We have incorporated the tables to the paper.
#2 There are a series of measures/scores that are not clear for a broader audience.
I suggest explaining the description of “M, t, p, Z, F(2), F, χ, F(1, 147), F(1, 157)” and how they are measured. The authors should at least name them explicitly.
For instance, the sentence
“(…) the scores referring to smartphone 185 addiction decreased from 33.42 ± 6,487 to 28.65 ± 7,968”.
How can a reader interpret this information? How is this score calculated? Is it counted in minutes or other quantities?
- We have edited the results section so that the scores are understandable to the reader. In addition, we have added two new columns to Table 1, one of which specifies the statistical analyzes done in each study.
#3 The authors should assist the reader in understanding their results. The paper is well-written, but in Section 3, it became exceedingly technical. I do not recommend removing any data or result, but explaining the metrics to the reader would improve the overall quality of this work.
Overall, I think the authors should define/explain/name each metric presented in Section 3.
- With the edition of the results section we have improved this aspect.
Recommendation
In my opinion, the scientific content of the paper is sound and the experimental results are convincing. The treatment of literature seems fair. Furthermore, I think that enough methodological details are provided for this work to be reproduced. Therefore, I recommend that this paper should be accepted for publication after minor revision.
- Thank you for all your suggestions and for your final comment.
* We have used the translation service to correct possible errors in grammar and spelling.
Reviewer 3 Report
It is not clear what is the novelty and contributions of the proposed work: does it propose a new method? Or does the novelty only consist in the application.
Author Response
It is not clear what is the novelty and contributions of the proposed work: does it propose a new method? Or does the novelty only consist in the application.
- Thank you for your comment. We have highlighted at the end of the introduction, not the novelty but the importance of the study, as well as its implications.
Over the past years, since the term “Internet addiction” first appeared in mass media and academic literature, research studies have gone a long way toward defining, exploring, investigating, describing, and predicting the phenomenon. A review of this research begs the following question: How should Internet addiction be treated? Can stopping all Internet usage cure Internet addiction? Is to stop using the Internet, to unplug or to throw out the computer the only way to cure Internet addiction?
With this study we want to break with the belief of "all or nothing" and expand the horizons of treatments.
* We have used the translation service to correct possible errors in grammar and spelling.
Reviewer 4 Report
I have been reading both the manuscript and the supplementary material, and due the volume of analysed studies, my initial recommendation is adding a column to the table in the supplementary material with the either the publication year of each study, or the year(s) when the studies were performed. As the triggers of these addictions are in part the different dynamics arisen from internet services, this annoation about the years is important from my point of view, as it could be relevant to understand the influence of several internet milestones, like rising popularity of this or that service in this or that geographic area (for instance TikTok / Douyin, Twitch, OnlyFans, ...), and loss of future effectiveness of this or that treatment.
I also suggest you to also include in the supplementary materials table columns for T0, T1, T2 and T3, and other relevant metrics, or a second table with these metrics. Although the information is already available in the manuscript's body, metrics in a table could help the readers to better understand the achievements and limitations of the explained studies and their proposed treatments.
A bit off topic, I'm wondering how difficult is going to be treating internet addictions and unhealthy human relationships after what human populations have suffered since 2020 due coronavirus pandemic outbreaks which led to severe restrictions at the human relationships level and alterations of social routines like physically meeting.
There is a repeated word at page 6, line 266, "which which".
Author Response
I have been reading both the manuscript and the supplementary material, and due the volume of analysed studies, my initial recommendation is adding a column to the table in the supplementary material with the either the publication year of each study, or the year(s) when the studies were performed. As the triggers of these addictions are in part the different dynamics arisen from internet services, this annoation about the years is important from my point of view, as it could be relevant to understand the influence of several internet milestones, like rising popularity of this or that service in this or that geographic area (for instance TikTok / Douyin, Twitch, OnlyFans, ...), and loss of future effectiveness of this or that treatment.
- Thank you very much for your observation. We have added a column with the years of each of the studies.
I also suggest you to also include in the supplementary materials table columns for T0, T1, T2 and T3, and other relevant metrics, or a second table with these metrics. Although the information is already available in the manuscript's body, metrics in a table could help the readers to better understand the achievements and limitations of the explained studies and their proposed treatments.
- We have incorporated a column into Table 1 where the statistical analyzes of each of the studies are collected. We have also edited the results section as it was not easy for the reader to understand. Now, we think that section is understandable and the reader can refer to it in the manuscript.
A bit off topic, I'm wondering how difficult is going to be treating internet addictions and unhealthy human relationships after what human populations have suffered since 2020 due coronavirus pandemic outbreaks which led to severe restrictions at the human relationships level and alterations of social routines like physically meeting.
- It is a very interesting comment and on which you have to start working. We have incorporated this observation in limitations, since in our study we wanted to analyze the programs that were operating before the pandemic. It is obvious that with the pandemic we must update ourselves in this regard, but perhaps we can build programs to intervene the misuse of the internet derived from the pandemic on the basis of many of the reviewed programs.
There is a repeated word at page 6, line 266, "which which".
- We have eliminated the repeated word.
Round 2
Reviewer 1 Report
The authors replied to the reviewer's comments and included in this version all the required modifications.
Author Response
Thank you for valuing our contributions.
Reviewer 3 Report
I thank the authors for considering my concern about the previous version of the manuscript. On some of the points I agree with the revisions made, and on other aspects I find them rather small revisions, and more work need to be done.
- In Section Introduction, you should highlight only the related works shortcomings and then provide general ideas on how your proposed approach addresses these issues and shortcomings.
2. The key technical contributions made in this work should be highlighted.
3. When the authors introduce the use of the Internet and technological devices, please consider into discuss the more related and recent works.
https://www.mdpi.com/2079-9292/10/22/2786
Author Response
Thank you for your comments.
1. We have emphasized in the introduction the contribution of our study.
3. We have tried to contrast the information with the most current references and related to our work.